# DentalArch: AI-Based Arch Shape Detection in Orthodontics

**J. D. Tamayo-Quintero** [1,2,*,†] **, J. B. Gómez-Mendoza** [1,†] **and S. V. Guevara-Pérez** [3,†]

1 Department of Electric, Electronic and Computing Engineering, Universidad Nacional de Colombia, Sede Manizales, Manizales 170003, Colombia; jbgomezm@unal.edu.co
2 Software Engineering Program, Faculty of Engineering, Tecnológico de Antioquia Institución Universitaria, Sede Robledo, Antioquia 050034, Colombia
3 Department of Oral Health-Orthodontics, Faculty of Dentistry, Universidad Nacional de Colombia, Sede Bogota, Bogota 050034, Colombia; svguevarap@unal.edu.co
* Correspondence: jdtamayoq@unal.edu.co
† These authors contributed equally to this work.

**Abstract:** Objective: This study aims to introduce and assess a novel AI-driven tool developed for the classification of orthodontic arch shapes into square, ovoid, and tapered categories. Methods: Between 2016 and 2019, we collected 450 digital dental models. Applying our inclusion and exclusion criteria, we refined our dataset to 50 models, ensuring a focused and detailed analysis. Plaster casts were digitized into 3D models with AutoScan-DS-EX. Three trained evaluators then measured mesiodistal and arch widths using MeshLab. The development of DentalArch was undertaken in two versions: the first version incorporates 18 input parameters, including mesiodistal widths (from the first molar to the first molar, totaling 14) and arch widths (1 intercanine, 2 interpremolar, and 1 intermolar, totaling 4); the second version uses only 4 parameters related to arch widths. Both versions aim to predict the arch shape. An evaluation of 28 machine learning methods through a k = 5-fold cross-validation was conducted to determine the most effective techniques. Results: In the tests, the performance evaluation of the DentalArch software in detecting arch shapes revealed that version 1, which analyzes 18 parameters, achieved an accuracy of 94.7% for the lower arch and 93% for the upper arch. The more streamlined version 2, which assesses only four parameters, also showed high precision with an accuracy of 93.0% for the lower arch and 92.7% for the upper arch. Conclusions: DentalArch provides a tool with potential use in orthodontic diagnostics, particularly in the task of arch shape classification. The software offers a less subjective and data-driven approach to arch shape determination. Moreover, the open-source nature of DentalArch ensures its global availability and encourages contributions from the orthodontic community.

**Keywords:** arch dental form; orthodontics; artificial intelligence; DentalArch software; AutoScan-DS-EX; machine learning; open source

## 1. Introduction

Since the inception of the term "computer-aided diagnosis" over six decades ago by physician Gwilym S. Lodwick [1] in a chest radiography study, artificial intelligence (AI) has revolutionized information processing. AI has emerged as a permanent fixture, enhancing human deductive reasoning and observational skills [2–4]. This technological breakthrough is grounded in emulating our cognitive processes across various domains, such as image processing and pattern recognition, and led to the development of Artificial Neural Networks (ANNs) [5]. ANNs have found extensive applications in medicine [6,7]. Significant advancements have been made in medical imaging [8,9], given the vast amount of information, storage requirements, and creation of image databases that need rapid and efficient analyses, enabling physicians to make decisions based upon the experience coded in such large datasets. In orthodontics and dentistry, one key assessment is the identification of the arch shape, as it provides a basis to the expectations that can be set during treatment and a path to carry it out.

Orthodontists and dentists utilize the arch shape as a tool to ascertain a patient's arch type. Preserving the arch shape or minimizing alterations during orthodontic treatments results in an improved stability during the retention phase. This was corroborated in 1995 by Cruz et al. [10], who conducted a long-term study on post-treatment dental arch shape changes, concluding that patients experiencing greater post-retention relapse have a direct correlation with an increased transversal magnitude [10]. Recent studies support this hypothesis, demonstrating higher relapse rates in arches that initially had a narrow shape and underwent increased transversal lengths [11,12].

Arch shapes can be categorized into three primary forms: square, ovoid, and tapered. This classification generally relies on the expert's subjective evaluation through human vision and/or other tools, such as millimetric templates employed for diagnostic examinations in orthodontics. However, this approach is susceptible to human error and observer bias—which have been evidenced and measured—leading to inaccurate or observer-dependent results [13,14].

Individual variations in arch configurations [15] pose challenges to the classification task. Several studies have tackled this problem using different approaches, including geometric analyses, mathematical shapes, and computerized methods to facilitate dental arch shape representation [16–18]. Various arch shapes have been proposed, such as the trifocal ellipse, catenary curve, parabola, U shape, modified sphere, and others [19,20]. Generally speaking, the dental arch shape and size exhibit significant diversity among different human groups, leading to the introduction of morphometric templates by orthodontists and commercial companies [21,22].

The accurate determination of the dental arch shape is important in orthodontics, greatly impacting the effectiveness and success of treatments [23]. This aspect is particularly significant in the context of utilizing memory wire arches, where the conformity of the wire to the patient's specific arch shape can dramatically affect the outcome of the treatment [24]. The compatibility between the arch shape and the applied corrective device is essential for achieving the desired dental alignment and occlusal harmony. Consequently, orthodontists rely on arch shape classification to tailor treatment plans that align with the unique dental structures of their patients. Innovations in diagnostic technologies, including the application of artificial intelligence for arch shape prediction, have markedly enhanced the accuracy and reliability of these classifications, promising improved treatment outcomes and patient satisfaction [25].

The intersection of artificial intelligence (AI) with orthodontic arch shape prediction has seen significant growth in interest and research [26]. Early methods for classifying arch shapes depended heavily on manual measurements and subjective evaluations, introducing considerable variability in results. However, the rise of AI and machine learning models has begun to change this landscape, offering innovative insights and methods for arch shape classification.

McKee and Molnar [27] utilized mathematical methods to categorize variations in dental arch shape among an Australian Aboriginal population, highlighting the diversity in arch forms and the challenges in their classification. Arai and Will [28] explored the correlation between subjective classifications of dental arch shape and objective analyses, shedding light on potential discrepancies and the need for more precise methodologies. Lee et al. [29] proposed a novel method to classify dental arch forms, employing clustering algorithms to sort arch shapes based on their goodness of fit and clinical applicability, demonstrating a shift towards more objective, data-driven classification methods. Takahashi et al. [30] developed an AI system using a convolutional neural network (CNN) to classify partially edentulous arches, marking a significant advancement in the AI-assisted design of removable partial dentures. Their system achieved high diagnostic accuracies, showcasing the potential of AI in dental arch classification. Qiu et al. [31] introduced DArch, a dental arch prior-assisted method for 3D tooth instance segmentation. By leveraging the dental arch structure, they enhanced the accuracy of tooth segmentation in 3D

dental models, emphasizing the value of incorporating dental arch priors into AI models for orthodontic applications.

In addition to these research efforts, commercial software solutions like Dolphin Imaging & Management Solutions® and Nemotec® have incorporated tools for dental arch analysis within their orthodontic planning and imaging suites [32]. While offering sophisticated imaging and diagnostic capabilities, it is important to note that these programs facilitate assisted rather than fully automatic arch shape detection. Practitioners use these tools to analyze and evaluate dental arches, but expert input is required to interpret and apply the software's findings in treatment planning. This distinction underscores the ongoing need for advancements in AI that can further automate and refine the process of arch shape classification and that aim to reduce the reliance on manual assessments and subjective evaluations.

These contributions collectively highlight the significant impact of AI on orthodontic arch shape classification and prediction, with "DentalArch" and similar AI-driven tools offering alternatives to traditional methods, alongside the assisted capabilities provided by existing commercial software.

The remainder of this document outlines the study structure, covering related work; materials and methods, encompassing data collection, arch form classification, software development, and machine learning technique evaluation; the main results attained with the tool; a brief discussion; and the principal conclusions drawn from our findings.

## 2. Materials and Methods

### 2.1. Data Collection

The dataset for this study initially comprised 451 complete dental cast models, both maxillary and mandibular, collected between 2016 and 2019. These models were sourced from patients at the Faculty of Dentistry, Universidad Nacional de Colombia, who sought dental care services. The fabrication of these models was obtained from impression taken with alginate and crafted from Type III and Type IV plaster. The original plaster models were obtained from a group of patients, all residents of Bogotá, who were of both sexes and were aged 18 years and older.

Following the application of the inclusion and exclusion criteria detailed in Table 1, the dataset was narrowed down to 50 models. This selection was not without its challenges, including label deterioration due to prolonged storage and the inherent fragility of the dental models, which occasionally resulted in a tooth loss or model disintegration, as depicted in Figure 1. These challenges underscore the complexities of assembling a high-quality dataset for dental research.

**Table 1.** Inclusion and exclusion criteria.

| Inclusion Criteria | Exclusion Criteria |
| --- | --- |
| Complete dentitions from the first molar to the first molar | Missing teeth within the maxilla or mandible |
| High-quality digital plaster models with clear visibility of all teeth | Incomplete or altered digital dental models (e.g., errors from scanning) |
| Legal adults aged 18 years and older | Restorations affecting the normal size and shape of teeth |
| | Congenital dental defects or tooth malformations |
| | Severe or moderate crowding |

This study received approval from the Ethics Committee of the Faculty of Dentistry at Universidad Nacional de Colombia on 28 September 2020 (Acta 20–20 B.CIEFO-154-2020).

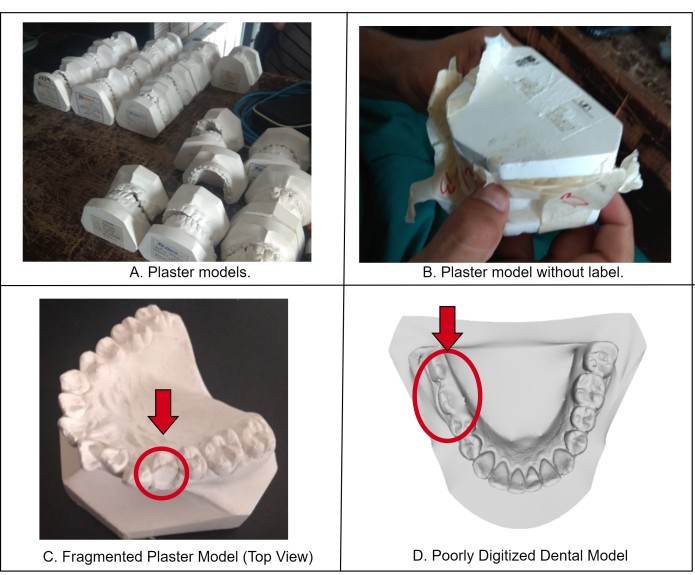

**Figure 1.** Challenges encountered during the model selection process, including label deterioration and model fragility. The red circles and arrows are used to indicate specific issues. In (**C**), the red circle indicates a plaster model where a piece of tooth was detached due to manipulation. In (**D**), the red circle highlights poor digitization.

*2.2. Arch Form Classification*

In the assessment of dental models, three orthodontic evaluators were enlisted, each with specific qualifications and levels of experience. These evaluators were responsible for measuring 19 parameters related to mesiodistal widths and arch shapes, which included 14 mesiodistal widths, 1 intercanine width, 2 interpremolar widths, 1 intermolar width, and the overall arch shape. Evaluator 1 and Evaluator 2 were residents specializing in orthodontics and maxillofacial orthopedics; Evaluator 3 was a general dentist. To ensure uniformity and dependability in their evaluations, all evaluators underwent calibration by an expert orthodontist before commencing this study. To establish the arch form, a consensus-based approach was implemented. The arch form classification was determined through a majority vote among the three evaluators, effectively mitigating the potential subjectivity inherent in individual assessments. This consensus approach not only ensured a comprehensive evaluation but also provided a robust and reliable reference for arch form classification.

MeshLab was developed by the ISTI-CNR research center and initiated at the University of Pisa in 2005 [33]. For the purposes of this study, the "measuring tool" feature of MeshLab was utilized to conduct manual assessments of 18 parameters on dental models, as illustrated in Figure 2. These measurements were carefully executed twice by the evaluators, with a fifteen-day interval between sessions, to guarantee the precision and reliability of the data collected. Furthermore, to enhance the consistency and validity of their assessments, the evaluators underwent specialized training conducted by an orthodontic expert prior to the commencement of this study.

The arch forms were categorized into three types: ovoid, square, and tapered. The classification criteria did not prescribe a minimum number of evaluators for consensus, but the aim of using three evaluators was to enhance the reliability of the classification process.

The chosen arch form classifications—ovoid, square, and tapered—were compatible with common clinical practice in orthodontics [34,35]. These shapes provide a practical and recognizable framework for describing variations in dental arch forms, facilitating communication among orthodontic professionals.

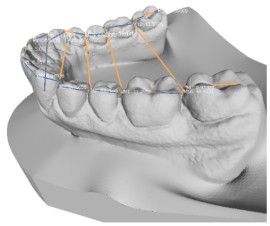
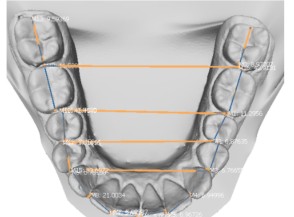

A. MeshLab Measurement Example:    B. Alternate View of MeshLab Measurements

**Figure 2.** MeshLab measurement examples: (**A**) shows the dental model with measurements captured in MeshLab, and (**B**) provides an alternate view of the same model with measurements. Note: The arch shape parameter was determined using conventional assessment methods.

*2.3. Tool Development*

The development of the DentalArch tool, tailored for arch shape detection, resulted in two distinctive versions:

- DentalArch v1: Incorporating a detailed set of 18 input parameters—comprising mesiodistal widths for each tooth from the first molar on one side to the first molar on the opposite side (14 parameters), along with widths for 1 intercanine, 2 interpremolars, and 1 intermolar—this version offers an exhaustive approach to analyzing arch shapes. This comprehensive parameter set facilitates a nuanced assessment of dental arches.
- DentalArch v2: This iteration simplifies the input to just 4 critical parameters— 1 intercanine width, 2 interpremolar widths, and 1 intermolar width. With a singular output parameter, DentalArch v2 simplifies the diagnostic process, emphasizing an ease of use and practicality without compromising the accuracy of its outputs. This version is designed for efficiency, making it particularly suitable for quick assessments while retaining the reliability expected of orthodontic diagnostic tools.

Both versions were developed using MATLAB® R2023B, ensuring a robust and efficient implementation. This choice of platform not only secures a high-performance tool solution but also promotes compatibility and seamless execution across diverse systems [36].

An essential feature of DentalArch is its user-friendly interface, allowing orthodontic professionals to manually input the necessary measurements. This intentional user-centric approach enhances the software's accessibility and ensures its seamless integration into the clinical practices of orthodontic professionals.

Furthermore, DentalArch exhibits Windows compatibility, being executable on any Windows-based computer, enabling orthodontic professionals to incorporate DentalArch into their practice without concerns about operating system constraints.

Machine Learning Techniques

In order to develop DentalArch, a comprehensive analysis of 29 different machine learning techniques was conducted. These techniques were systematically evaluated to determine the most suitable method for the task of classifying dental arch shapes. The selection process aimed to identify a technique that could provide high accuracy in this classification task.

The machine learning techniques were categorized into various groups, including decision trees, discriminant analysis, naive Bayes classifiers, support vector machines, nearest neighbor classifiers, ensemble classifiers, and neural network classifiers. These categories were chosen to encompass a wide range of methodologies and capture diverse approaches to classification.

The performance of each technique was assessed based on its accuracy in classifying dental arch shapes. By comparing the results obtained from these techniques, we were able to identify the most effective approach for the development of DentalArch.

Table 2 provides a summary of the machine learning techniques analyzed in this study, along with their respective categories.

**Table 2.** List of machine learning techniques tested in this work.

| Category | Techniques |
|---|---|
| Decision trees | Fine tree<br>Medium tree<br>Coarse tree |
| Discriminant analysis | Linear discriminant<br>Quadratic discriminant |
| Naive Bayes classifiers | Gaussian naive Bayes<br>Kernel naive Bayes |
| Support vector machines | Linear SVM<br>Quadratic SVM<br>Cubic SVM<br>Fine Gaussian SVM<br>Medium Gaussian SVM<br>Coarse Gaussian SVM |
| Nearest neighbor classifiers | Fine KNN<br>Medium KNN<br>Coarse KNN<br>Cosine KNN<br>Cubic KNN<br>Weighted KNN |
| Ensemble classifiers | Boosted trees<br>Bagged trees<br>Subspace discriminant<br>Subspace KNN<br>RUSBoost trees |
| Neural network classifiers | Narrow neural network<br>Medium neural network<br>Wide neural network<br>Bilayered neural network<br>Trilayered neural network |

*2.4. Data Augmentation and Dataset Composition*

In the process of training the DentalArch software, a total of 200 data points with corresponding labels were utilized. These data points were derived from meticulous manual evaluations conducted by orthodontic professionals, including their manual measurements.

To enhance the robustness of the dataset and facilitate more effective model training, data augmentation techniques were employed. Specifically, measurements of the same dental models at different time points were incorporated as part of the augmentation process. This temporal variation in the dataset allowed for the representation of natural changes that may occur over time, contributing to a more dynamic and comprehensive training set.

The decision to refrain from introducing synthetic data was made to prevent any potential contamination of the dataset and to maintain a high level of relevance to the clinical context [37]. All data points used in the training process accurately represented the variations and complexities encountered in actual orthodontic practice.

Furthermore, the augmentation process, utilizing measurements from different time points, enabled the expansion of the dataset without compromising its authenticity [38]. This approach contributed to a more comprehensive and diverse set of training examples, reflecting the dynamic nature of dental arch shapes over time.

By leveraging these carefully collected and temporally augmented datasets, DentalArch was trained on a rich and varied array of real-world scenarios, enhancing its ability to generalize and make accurate predictions in a clinical setting.

### 2.4.1. Model Selection and Implementation

After evaluating the machine learning techniques, the method with the highest accuracy was selected for the implementation of DentalArch.

### 2.4.2. Model Validation

To validate the developed DentalArch software, we employed a k-fold cross-validation with a value of k equal to 5. This approach was chosen to rigorously assess the model's performance while preventing overfitting and ensuring its generalizability. In k-fold cross-validation, the dataset is divided into five distinct subsets, or "folds". The model is trained on four of these folds and tested on the remaining one in each iteration. This process is repeated five times, with a different fold serving as the test set in each iteration.

While k-fold cross-validation is computationally demanding, it offers several advantages. It maximizes the use of available data, making it particularly valuable when dealing with limited sample sizes. By evaluating the model's performance across different subsets of data, we obtain a robust measure of its effectiveness and its ability to generalize to new, unseen data.

### 2.5. Performance Metric

The performance metric for measuring DentalArch's effectiveness was accuracy. The method with the highest precision was chosen for the development of the tool.

### 2.6. Statistical Analysis

To ensure the accuracy of arch shape classifications in our study, we used Minitab® 21.4, a data analysis software suite developed at Pennsylvania State University [39]. Within Minitab® 21.4, the Cohen's kappa test was applied to statistically assess the agreement among evaluators on arch shapes and our method at a 5% significance level. This approach ensured that our classifications were consistent and reliable, with the standard for classification established through a consensus method known as weighted majority voting, where the most common classification among evaluators was adopted as the criterion. Moreover, to evaluate the reliability of our measurements and the consistency of assessments made by our evaluators, a Gauge Repeatability and Reproducibility (Gage R&R) analysis was performed.

### 2.7. Upper and Lower Arch Analysis

The maxillary (upper) and mandibular (lower) arches were analyzed independently. Consequently, separate algorithms were developed for the upper and lower arches in each version of DentalArch. The efficacy of both versions was assessed by a third-year resident from an orthodontics and maxillofacial orthopedics program.

## 3. Results

### 3.1. Repeatability and Reproducibility

The Gage R&R analysis from our study, which focused on arch widths, revealed the following key findings: Total Gage R&R variance contributions ranged from 4.74% to 9.20%, indicating the extent of the measurement system variability. Repeatability errors were minimal, with percentages between 0.58% and 1.22%, suggesting consistent measurements across repeated evaluations. Reproducibility errors, reflecting the evaluators' variability and their interaction with the models, were more significant, contributing 1.37% to 8.21% to the total variance.

### 3.2. Evaluating Performance of Machine Learning Techniques across Dental Arches

In this study, we evaluate the efficiency of various machine learning models applied to the analysis of dental arches, specifically DentalArch v1 and v2. A diverse set of models was employed, the performance of each being gauged in terms of accuracy. Table 3 encapsulates the average accuracy achieved by each model.

**Table 3.** Accuracy of machine learning techniques for DentalArch.

| Method | Models | Dv1_Low | Dv2_Low | Dv1_Up | Dv2_Up | Avr. |
|---|---|---|---|---|---|---|
| Decision trees | 1 Fine tree | 80.7 | 78.3 | 79.7 | 80.3 | 79.7 |
| | 2 Medium tree | 80.7 | 77.7 | 80 | 80.7 | 79.7 |
| | 3 Coarse tree | 81 | 87 | 79.7 | 82 | 81.1 |
| Discriminant analysis | 4 Linear discriminant | 82.7 | 76.7 | 77.3 | 77 | 78.4 |
| | 5 Quadratic discriminant | Failure | 78.7 | Failure | 82.3 | 80.5 |
| Naive Bayes classifiers | 6 Gaussian naive Bayes | 72.7 | 65 | 64.7 | 68.3 | 67.6 |
| | 7 Kernel naive Bayes | 78.7 | 73 | 73.7 | 78.7 | 75.6 |
| Support vector machines | 8 Linear SVM | 81 | 76 | 76.3 | 76 | 77.3 |
| | 9 Quadratic SVM | 90.3 | 82 | 87.7 | 85.3 | 86.3 |
| | 10 Cubic SVM | 90.7 | 82.7 | 87.3 | 85 | 86.4 |
| | 11 Fine Gaussian SVM | 78.7 | 86.3 | 78.3 | 91 | 83.5 |
| | 12 Medium Gaussian SVM | 85.3 | 79.3 | 80.7 | 82.3 | 81.9 |
| | 13 Coarse Gaussian SVM | 76 | 76 | 76 | 76 | 76 |
| Nearest neighbor classifiers | 14 Fine KNN | 92.3 | 93 | 90.7 | 92.3 | 91.6 |
| | 15 Medium KNN | 79.7 | 77 | 78.3 | 84 | 79.7 |
| | 16 Coarse KNN | 76 | 76 | 76 | 76 | 76 |
| | 17 Cosine KNN | 82.3 | 74.7 | 75 | 83 | 78.3 |
| | 18 Cubic KNN | 79.3 | 77 | 78.3 | 83.7 | 79.5 |
| | 19 Weighted KNN | 88.7 | 88.7 | 84 | 92.7 | 88.5 |
| Ensemble classifiers | 20 Boosted trees | 82 | 86.3 | 82 | 83.3 | 83.4 |
| | 21 Bagged trees | 87.7 | 87 | 84.3 | 87.3 | 86.5 |
| | 22 Subspace discriminant | 79.3 | 76 | 76 | 75.3 | 76.6 |
| | 23 Subspace KNN | 94.7 | 86 | 93 | 87.7 | 90.3 |
| | 24 RUSBoost trees | 76 | 69.3 | 56.3 | 72.7 | 68.5 |
| Neural network classifiers | 25 Narrow neural network | 88.3 | 86 | 84 | 83.3 | 85.4 |
| | 26 Medium neural network | 87.3 | 86 | 84.7 | 86.7 | 86.1 |
| | 27 Wide neural network | 88.3 | 85 | 87.7 | 88.7 | 87.4 |
| | 28 Bilayered neural network | 89.7 | 85 | 83.7 | 88.3 | 86.6 |
| | 29 Trilayered neural network | 84 | 82.7 | 77.3 | 86.3 | 82.5 |

### 3.2.1. Lower Jaw

We provide a comparative analysis of DentalArch v1 and v2 software in their application for lower jaw arch prediction. Visual representations of each software in action can be observed in Figures 3 and 4, showcasing the user interface and the graphical prediction of the arch shape.

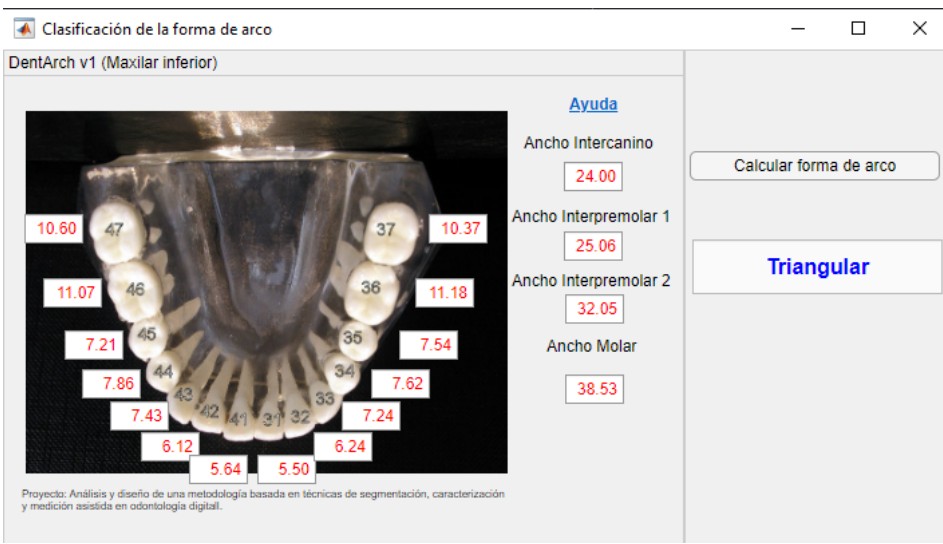

**Figure 3.** Screenshot of DentalArch v1 in action. The image illustrates how the software generates the predicted lower jaw arch shape.

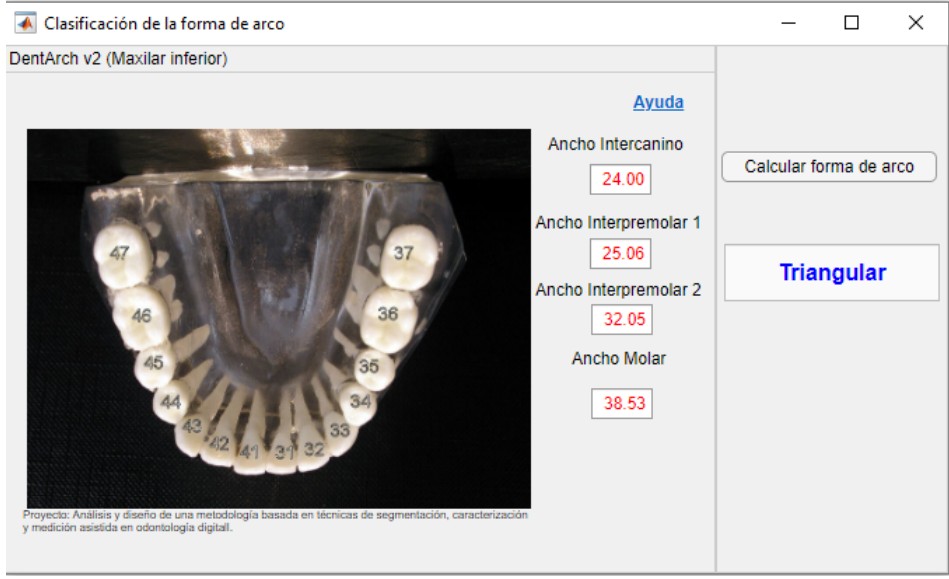

**Figure 4.** Screenshot of DentalArch v2 in action. Similar to v1, the software graphically presents the predicted lower jaw arch shape. However, due to its reduced complexity, the v2 interface is more user-friendly.

In the case of the lower jaw arch, DentalArch v1 successfully identified 48 out of 50 models. Specific values corresponding to various arch shapes—square, ovoid, and tapered—are available in Table 4. An additional validation, using 20% data unknown to the model, resulted in a 90% accuracy rate, within a confidence interval of (55.50–99.75). The agreement indices for these unknown data are also presented in Table 5.

**Table 4.** Concordance indices for different arch shapes and unknown data with DentalArch v1.

| Response | Known Data | | | Unknown Data | | |
|---|---|---|---|---|---|---|
| | **Kappa** | **SE Kappa** | **Z** | **Kappa** | **SE Kappa** | **Z** |
| Square | 1.00000 | 0.141421 | 7.07107 | 0.73333 | 0.316228 | 2.31900 |
| Ovoid | 0.90099 | 0.140726 | 6.40242 | 1.00000 | 0.316228 | 3.16228 |
| Tapered | 0.77876 | 0.137917 | 5.64660 | 0.73333 | 0.316228 | 2.31900 |
| Overall | 0.90909 | 0.108567 | 8.37355 | 0.84000 | 0.228035 | 3.68364 |

**Table 5.** Concordance indices for different arch shapes and unknown data with DentalArch v2.

| Response | Known Data | | | Unknown Data | | |
|---|---|---|---|---|---|---|
| | **Kappa** | **SE Kappa** | **Z** | **Kappa** | **SE Kappa** | **Z** |
| Square | 0.787385 | 0.141421 | 5.56765 | 1.00000 | 0.316228 | 3.65643 |
| Ovoid | 0.890351 | 0.141421 | 6.29573 | 0.733333 | 0.316228 | 2.31343 |
| Tapered | 0.846390 | 0.141421 | 5.98488 | 0.60784 | 0.316228 | 1.91900 |
| Overall | 0.845600 | 0.113195 | 7.47027 | 0.753090 | 0.242586 | 3.10422 |

In Table 4, we display the kappa statistic, its standard error (SE kappa), and the Z-score for both known and unknown data across the three dental arch shapes—square, ovoid, and tapered—as well as overall. This comparative layout enables us to evaluate the performance of DentalArch v1 on known and unknown data concurrently.

As for DentalArch v2, it managed to correctly identify 47 out of 50 models using known data and 9 out of 10 models with unknown data. The corresponding concordance indices for the three arch shapes—square, ovoid, and tapered—are documented in Table 5.

### 3.2.2. Upper Jaw

The performance of the DentalArch v1 and v2 software for the upper jaw arch was also evaluated. Screenshots of the software while processing an upper jaw sample are illustrated in Figures 5 and 6. These figures showcase the user interface and graphical representation of the predicted arch shape for the upper jaw.

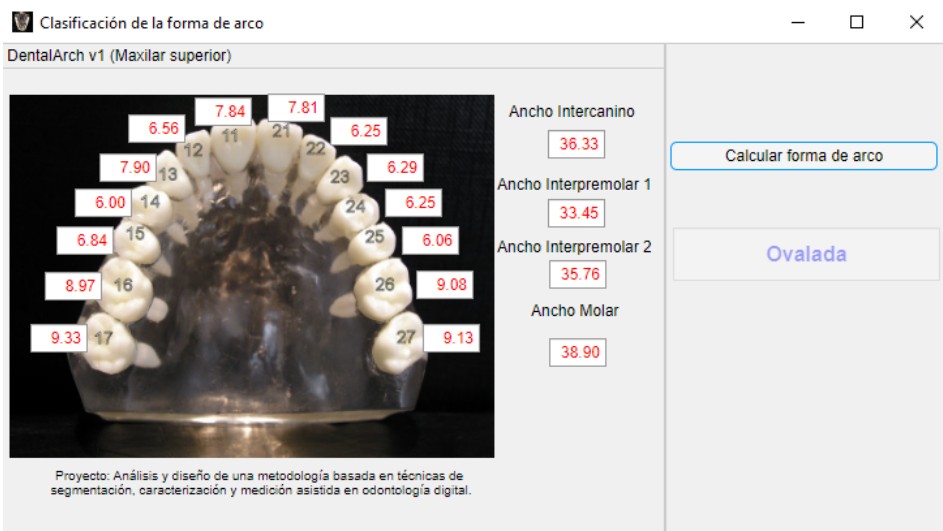

**Figure 5.** Screenshot of DentalArch v1 analyzing an upper jaw sample. The image showcases how the software generates the predicted upper jaw arch shape.

For the upper jaw arch, DentalArch v1 correctly identified 49 out of 50 models. The specific values for the different arch shapes, square, ovoid, and tapered, are documented in Table 6. An additional validation was conducted with 20% data unknown to the model, achieving an 80% accuracy rate. The agreement indices for these unknown data are listed in Table 6.

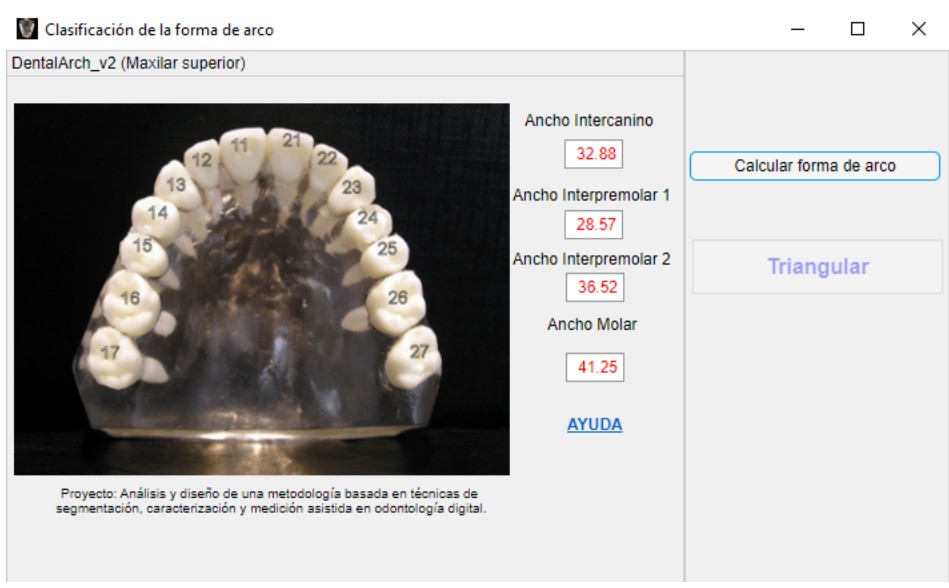

**Figure 6.** Screenshot of DentalArch v2 processing an upper jaw sample. Analogous to v1, the software graphically presents the predicted upper jaw arch shape. The v2 interface, with its reduced complexity, promotes ease of use.

**Table 6.** Concordance indices for different arch shapes and unknown data with DentalArch v1 for the upper jaw.

| Response | Known Data | | | Unknown Data | | |
|---|---|---|---|---|---|---|
| | **Kappa** | **SE Kappa** | **Z** | **Kappa** | **SE Kappa** | **Z** |
| Square | 0.95636 | 0.152933 | 6.24605 | 0.78034 | 0.31624 | 2.46830 |
| Ovoid | 0.91376 | 0.151529 | 6.02548 | 0.88923 | 0.31623 | 2.81214 |
| Tapered | 0.87183 | 0.146396 | 5.95671 | 0.71198 | 0.31624 | 2.25050 |
| Overall | 0.91372 | 0.123631 | 7.39016 | 0.79385 | 0.232523 | 3.41529 |

The DentalArch v2 software correctly identified 46 out of 50 models with known data and 7 out of 10 models with unknown data. The concordance indices for different arch shapes, square, oval, and tapered, are detailed in Table 7.

**Table 7.** Concordance indices for different arch shapes and unknown data with DentalArch v2 for the upper jaw.

| Response | Known Data | | | Unknown Data | | |
|---|---|---|---|---|---|---|
| | **Kappa** | **SE Kappa** | **Z** | **Kappa** | **SE Kappa** | **Z** |
| Square | 0.75100 | 0.141421 | 5.30952 | 0.45600 | 0.316228 | 1.44169 |
| Ovoid | 0.85776 | 0.141421 | 6.06253 | 0.67770 | 0.316228 | 2.14259 |
| Tapered | 0.81195 | 0.141421 | 5.74038 | 0.61934 | 0.316228 | 1.95807 |
| Overall | 0.80690 | 0.118465 | 6.80569 | 0.58435 | 0.226550 | 2.57800 |

This side-by-side comparison provides a comprehensive view of the performance of both software versions for the upper jaw arch.

### 3.3. Computational Tool Development

Two versions of the computational tool were developed for this study:

- DentalArch v1: 18 parameters, including 14 mesiodistal widths, 1 intercanine width, 1 intermolar width, and 2 interpremolar widths (accuracy: 94.7% lower arch, 93% upper arch).
- DentalArch v2: A reduced version with only four parameters, including one intercanine width, one intermolar width, and two interpremolar widths. This version

provides a faster prognosis but with a slightly lower precision (accuracy: 93.0% lower arch (−1.7%), 92.7% upper arch (−0.3%)).

## 4. Discussion

Integrating AI-driven tools like DentalArch or similar tools into orthodontic practice for dental arch shape classification could offer significant enhancements. These tools not only ease the diagnostic process but also achieve precision and efficiency comparable to expert evaluations while reducing subjectivity. This is particularly valuable given the strong link between the dental arch shape and individual growth patterns [40,41]. By accurately assessing the arch shape, these tools can ensure treatment plans are aligned with each patient's unique developmental trajectory. Building upon this, the shift towards digital models in the field represents a paradigm shift, offering substantial savings in both time and resources. Digital models have emerged as a clinically viable alternative, embracing the digital transformation that encompasses computational intelligence, augmented reality, and three-dimensional printing technologies [42].

From the tests carried out with our data, DentalArch version 1 attained a 94.7% accuracy detecting the arch shape for the lower arch and a 93% accuracy for the upper arch. The more streamlined version 2, which assesses only four parameters, achieved an accuracy of 93.0% for the lower arch and 92.7% for the upper arch. These results are in line with recent trends towards data-driven diagnostics in orthodontics, which were addressed by McKee and Molnar [27] and Arai and Will [28]. The use of machine learning, as explored in research by Takahashi et al. [30] and Qiu et al. [31], underscores the potential for computational techniques to enhance diagnostic precision and reduce observer bias.

When compared to established orthodontic software solutions like Dolphin Imaging & Management Solutions® and Nemotec®, DentalArch offers a novel, semiautomated approach to arch shape classification. Unlike these traditional platforms, which may require manual input for detailed analyses, DentalArch leverages AI to provide objective classifications with minimal user intervention. Furthermore, its open-source nature stands in contrast to the proprietary models of existing software, encouraging innovation and allowing for broader adaptation and improvement within the orthodontic community.

DentalArch v2 relies on just four key input parameters for determining the dental arch form. This refinement stems from recent research emphasizing the critical role of specific arch measurements, particularly the intercanine width, in predicting dental arch shapes. Abdul Rehman et al. (2021) highlighted the strong link between the intercanine width and the forms of mandibular dental arches, validating the approach of using a subset of measurements for effective orthodontic analysis [43]. Motivated by such insights, DentalArch v2 was designed to enhance diagnostic precision by focusing on essential measurements like the intercanine width (ICW) and intermolar width (IMW), corroborated by research [44,45], demonstrating their significant correlation with the overall arch form. This led to a substantial reduction in input parameters, simplifying the diagnostic workflow without compromising the prediction accuracy of dental arch shapes.

The comprehensive evaluation of 29 machine learning methodologies for the development of DentalArch illustrates a strategic approach to enhancing orthodontic diagnostic tools. This analysis, which spanned from decision trees to neural network classifiers, was not only pivotal in identifying the most effective algorithms for arch shape prediction but also in maintaining the high diagnostic accuracy of DentalArch v2 despite its simplified model.

The development of DentalArch, with an aim to apply it to orthodontic diagnostics, successfully balanced detailed accuracy and user convenience across lower and upper jaw analyses. While DentalArch v1's extensive parameters slightly edged out v2, the latter offered a compelling advantage, marking a minor compromise for significantly improved accessibility.

The preference for utilizing the concordance index, specifically the kappa coefficient, over sensitivity in assessing DentalArch was motivated by the complexity of the evaluation

task, which encompassed multiple categories of arch shapes. Concordance indices, such as the kappa coefficient, are commonly applied in studies evaluating agreement between different classification methods or observers, particularly in scenarios with more than two potential categories. This decision was rooted in the necessity to evaluate not only the accuracy of classification within each specific category but also the overall agreement between the classifications and the actual categories while considering chance agreement. Additionally, the kappa coefficient's ability to adjust observed concordance for chance agreement was instrumental in providing a measure of agreement beyond random chance. Furthermore, the work by Landis and Koch (1977) emphasizes the relevance of the kappa coefficient in evaluating agreement between classifications of categorical data, especially in contexts with multiple categories [46].

Despite its promising utility in identifying and classifying dental arch shapes, this study acknowledges certain limitations, including a relatively small sample size, potential biases in evaluator evaluations, and the absence of extensive real-world testing. Validation by a single orthodontics and maxillofacial orthopedics resident may not provide a comprehensive evaluation.

Ongoing efforts aim to address these limitations through further testing in larger clinical settings. This study's implications extend to the field of dental research, emphasizing potential advantages in utilizing artificial intelligence for dental arch shape classification. Rigorous validation within actual clinical contexts remains a crucial focus for future explorations.

## 5. Conclusions

DentalArch provides a tool with potential use in orthodontic diagnostics, particularly in the task of arch shape classification. The tool offers a less subjective and data-driven approach to arch shape determination. Moreover, the open-source nature of DentalArch ensures its global availability and encourages contributions from the orthodontic community. However, it is essential to acknowledge this study's limitations, including a relatively small sample size, which may affect the generalizability of these findings.

DentalArch's evolution from version 1 to version 2 underscores the potential to enhance the diagnostic precision and user experience in orthodontic practice. However, it is important to interpret these results cautiously, recognizing that DentalArch serves as a supportive tool rather than a definitive solution. Further validation across diverse patient groups and clinical conditions is necessary to refine its algorithms and ensure its reliable applicability in clinical practice.

Future research should aim to address these limitations by expanding this study's scope to include a wider array of scenarios, thereby enhancing the robustness and reliability of DentalArch. Integrating the tool into a broader spectrum of orthodontic and dental care practices might offer valuable support in treatment planning and outcomes, promoting more personalized and efficient patient care.

**Author Contributions:** Each author made significant contributions to the conception, design, and execution of this research study. The specific contributions of each author are as follows. J.D.T.-Q.: conceptualization, programming, methodology, writing—original draft, and data curation. J.B.G.-M.: writing—reviewing and editing, methodology, and supervision. S.V.G.-P. formal analysis, resources, data curation, and writing—reviewing and editing. All authors have read and agreed to the published version of the manuscript.

**Funding:** We express our sincere gratitude to Colciencias for their crucial support provided through the Convocatoria No. 757 de 2016 scholarship. This support was instrumental in facilitating the participation of J.D. Tamayo-Quintero, contributing significantly to the success of this research.

**Institutional Review Board Statement:** We are pleased to provide the Institutional Review Board (IRB) Statement and approval details relevant to our study. This study was conducted in strict accordance with the principles outlined in the Declaration of Helsinki. Approval for this research was obtained from the Ethics Committee of the Faculty of Dentistry at Universidad Nacional de Colombia

on 28 September 2020, with protocol code Acta 20–20 B.CIEFO-154-2020. This approval underscores our commitment to ethical standards in conducting studies involving human participants.

**Informed Consent Statement:** In compliance with publication standards, we would like to state that written informed consent for participation in this study was not applicable in this case. The research involved retrospective analysis based on gypsum models, and the subjects involved were part of the university setting where patients provide consent at the outset for educational and research purposes. It is important to note that ethical considerations were rigorously maintained, adhering to the university's policies and protocols.

**Data Availability Statement:** The data supporting the reported results of this study are available upon request. Due to privacy and ethical restrictions, the data are not publicly archived. Researchers interested in accessing or analyzing the data may contact the corresponding author, Dr. Juan Tamayo, at the email address jdtamayoq@unal.edu.co. Further details and inquiries regarding data availability and access will be promptly addressed by the corresponding author.

**Acknowledgments:** In acknowledging the contributions and support received during the course of this research, we extend our gratitude to Colciencias now MinCiencias for their vital support through the Crédito educativo condonable, Convocatoria No. 757 de 2016, enabling J.D. Tamayo-Quintero's participation and greatly contributing to the success of this study. Furthermore, we express our deep appreciation to the Universidad Nacional de Colombia for their invaluable collaboration and resources throughout the research process. Their support has been pivotal in the accomplishment of our objectives. Our sincere thanks are also extended to the experts and individuals who generously shared their time and expertise, significantly enhancing the quality and depth of our study. Their valuable contributions have left a lasting impact on our research. These acknowledgments signify our appreciation for the collective efforts and support that have shaped the outcomes of this research project.

**Conflicts of Interest:** The authors declare no conflicts of interest. The funders had no role in the design of the study; in the collection, analyses, or interpretation of data; in the writing of the manuscript; or in the decision to publish the results.

## Abbreviations

The following abbreviations are used in this manuscript:

| | |
|---|---|
| MDPI | Multidisciplinary Digital Publishing Institute |
| AI | Artificial Intelligence |
| ANNs | Artificial Neural Networks |
| kNN | k-Nearest Neighbors |
| SVM | Support Vector Machines |
| SE Kappa | The Standard Error for an Estimated Kappa |
| IRB | Institutional Review Board |

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
