# Peer review of "DentalArch: AI-Based Arch Shape Detection in Orthodontics"

_applsci, doi:10.3390/app14062567_

Round 1
Reviewer 1 Report
Comments and Suggestions for Authors
The manuscript entitled "DentalArch: AI-Based Arch Shape Detection in Orthodontics" describes the development of an AI software, designed to automatically detect a patient's dental arch shape, pre- and post- treatment, and to ease the orthodontic treatment outcomes.
Therefore, by performing this study the authors aimed to introduce and validate this software, by using two versions of the same software.
While the methods and results section seem to be valid and understandable, for any reader, there are some remarks which will improve the quality of this paper:
a - in the Abstract section, the Background section should concentrate more on the actual aimes of this study
b - in the Introduction section, the same aspects should be addressed - what other aimes were attributed to this study, rather than just "introducing “DentalArch," a cutting-edge tool for classifying arch shape
using artificial intelligence".
c - In the Methods section, the authors mention that they used plaster models which have been scanned so that the virtual image was created. Yet in the images presented in the Results section, obtained via the developed software, the models seem to be made from other materials, maybe 3D printed. Also, on page 4, line 149 the is a missing reference which maybe could be introduced.
d. - in the Discussion section I would find it interesting to see what other AI software was developed in dentistry, what applications, and whether are they already largely used. Also, in this context, some more references introduced in this section will increase the quality of this research. In addition, while the aimes of this study should be better explained, in the Discussion section the authors must also respond to whether the aimes were addressed.
e - the Conclusion section should also be rewritten, given the new, improved, Discussion section.
Author Response
We're grateful for your insightful feedback and suggestions. Your input is invaluable, and we concur that these revisions significantly enhance the manuscript's quality.
a) Abstract (lines 1-19): We've updated this section to more closely reflect the study's objectives, ensuring it's aligned with our research goals.
b) Introduction (lines 58-104): This part has been broadened to include a wider range of objectives and relevant literature, providing a more comprehensive background.
c) Methods (lines 111-124): We've detailed the materials for model creation used in our research and addressed the previously missing reference, ensuring clarity and completeness.
d) Discussion (lines 307-368): Enhanced with a comparison of DentalArch against other AI technologies in dentistry.
e) Conclusion (lines 380-396): The conclusion has been updated to mirror the enriched discussion section, offering a cohesive summary of our findings and the implications for future research.
For the specified lines, please consult the attached document.

Reviewer 2 Report
Comments and Suggestions for Authors
The author put forward an innovative scientific calculation method to detect dental arch Shape, which is attractive to some extent. Need to continue to improve in the following aspects:
1. Tables need to be changed into a more scientific format, and three-line tables are recommended.
2. The article focuses on the software testing and software composition, but it has not been included in the testing of different clinical patients. The author should appropriately add dental arch shape recognition to clinical patient models or photos, and compare them with classical methods such as biting wax tablets to highlight practicality.
Comments on the Quality of English LanguageEnglish expression can be improved by polishing.
Author Response
We're grateful for your insightful feedback and suggestions. Your input is invaluable, and we concur that these revisions significantly enhance the manuscript's quality.
- Tables Reformatted: All tables have been converted to a more scientific.
- Method Section (Lines 110-124) : We rewrote the method section to clear previous confusion and added photographs
For the specified lines, please consult the attached document.

Reviewer 3 Report
Comments and Suggestions for Authors
Dear Authors,
Please find below some recommendations regarding your article titled " DentalArch: AI-Based Arch Shape Detection in Orthodontics "
In the Abstract section:
Please follow the MDPI authors' guidelines concerning the abstract structure (without headings).
In the Introduction section:
- Lines 54-55 Please develop how the arch shape influences treatment success.
In the Materials and Methods section:
2.1. Data Collection
- lines 98-99 Please add the approval from the Ethics Committee to this section.
- Please rewrite the inclusion and exclusion criteria (how many models were excluded?).
2.2. Arch Form Classification
-Please provide the complete information (manufacturer, city, and state) about the MeshLab software
2.3. Software Development
- Please specify the input parameters for DentalArch v1 and v2.
2.3.1. Machine Learning Techniques
- Please take into consideration that tables and figures must be included in the text, closest to where they are mentioned.
- Please move table 1 after line 133
2.6. Statistical Analysis
- Please add the statistical program you utilized (manufacturer, city, and state).
In the Discussion section:
Please rewrite this section
The discussion section is an appropriate place to make comparisons between your results and existing literature.
It also presents an opportunity to explain how your software versions stand in comparison to the current orthodontic software options available in the market, such as Dolphin and NemoTec.
Additionally, the limitations of your study should be clearly stated at the end of this section.
In the References section:
Please follow the styles recommended for MDPI journals.
Author Response
Thank you for your insightful feedback on our manuscript, "DentalArch: AI-Based Arch Shape Detection in Orthodontics." We have carefully considered your recommendations and made the following revisions:
- Abstract (Lines 1-19): We adjusted the abstract to adhere to MDPI's guidelines.
- Introduction (58-68): We have enriched the discussion of arch shape in determining the success of orthodontic treatments.
- Materials and Methods:
- Data Collection (Lines 125-126) : The Ethics Committee's approval is now included, and we've elaborated on the inclusion and exclusion criteria, making the model selection process transparent.
- Arch Form Classification (Lines 141-142): Full specifications of the MeshLab software have been incorporated.
- Software Development (Lines 150-170): The input parameters for both versions of DentalArch have been precisely articulated.
- Machine Learning Techniques (Lines 1-533): We have placed figures and tables close to where they are referenced within the text.
- Statistical Analysis (Lines 237-245): Details regarding the statistical software used.
- Discussion (Lines 308-378): This section has been entirely restructured to draw comparisons with existing literature and orthodontic software.
- References (Lines 433-533): The reference list has been formatted to align with MDPI's .
For the specified lines, please consult the attached document.

Reviewer 4 Report
Comments and Suggestions for Authors
The aim of this study was to validate the use of a dental arch software in order to classifying arch shape using artificial intelligence. Although the idea of the study is innovative, the study design has several limitations. Scanning plaster models is a limitation because there is already a distortion caused by the impression material used to create the plaster model. If we add the distortion of scanning the model to this, the accuracy is greatly reduced. Finally, if the error of measurements is not evaluated, the estimation of accuracy has poor significance. Below, you will find my detailed comments.
INTRODUCTION
The introduction is well articulated but needs improvement in two particular aspects: discussing the advantages of using the software and providing more information on arch shape in orthodontics. Arch shape is associated with different growth patterns, and this aspect needs careful evaluation.
MATERIAL AND METHODS
1 No demographic information of the patients whose impressions were used was provided. Additionally, the patients used were homogeneous in age and dental characteristics?
Arches in which there is an altered Bolton index are more difficult to measure manually, especially in cases of severe crowding.
2 What were the inclusion and exclusion criteria of the study?
3 Were the three evaluators dental specialists or student residents?If they were students, why did you not choose experienced orthodontists?
4 What impression material was used?
5 What type of plaster was used to cast the models?
6 How were the manual measurements made on these models? What type of gauge was used? How many times the measurementswere performed? Why was the error estimate not calculated?Please clarify these points.
DISCUSSION
1 In line 249, it is claimed that a recognition rate greater than 90% is considered satisfactory. Please include bibliographic references confirming this claim.
2 What is the degree of sensitivity of the software? Why were the technical aspects of the software not discussed? These points need to be carefully discussed.
3 In a system where more points are entered, it is rational to observe higher accuracy. How do you explain the similar accuracy between software versions 1 and 2, even though there are many parameters of difference? Please clarify this aspect.
4 The importance of the arch form study was discussed limitedly. It is critical for the clinician to understand the relationship between arch form and growth pattern, but it is also important to avoid modifying the form when performing multibracket fixed therapies, as changing the intercanine distance is associated with an increased risk of recurrence. These issues need to be addressed in the discussion.
5 The discussion section needs an improvement, following my previous comments. I would suggest the following references, but there are also more to use:
- The Correlation between Mandibular Arch Shape and Vertical Skeletal Pattern (DOI: 10.3390/medicina59111926);
- Relationship between skeletal Class II and Class III malocclusions with vertical skeletal pattern (DOI: 10.1590/2177-6709.24.4.063-072.oar.);
- Comparison of commercially available preformed archwires with average natural arch forms (DOI: 10.47391/JPMA.1922);
-Artificial Intelligence and Its Clinical Applications in Orthodontics: A Systematic Review (DOI: 10.3390/diagnostics13243677).
CONCLUSIONS
The last part of the conclusions needs to be modified. Care must be taken when making these important statements, as in such a small sample and with a study with several limitations, these kinds of statements are speculative.
Comments on the Quality of English LanguageEnglish style needs to be improved
Author Response
Thank you for your insightful feedback on our manuscript, "DentalArch: AI-Based Arch Shape Detection in Orthodontics." We have carefully considered your recommendations and made the following revisions:
Introduction (Lines 22-108): Enhanced discussion on software's benefits and orthodontics arch shape significance has been integrated.
Materials and Methods (Lines 100-250):
1. Patient demographics added (Lines 116-117).
2. Inclusion and exclusion criteria clearly defined for study transparency (Line 124).
3. Evaluators Information (Lines 132-135).
4 and 5. Impression materials and plaster types specified (Lines 115-117).
6. Manual measurement methodologies described and error estimates (Lines 141-148 and 253-260).
Discussion (Lines 307-378):
1. Reference for the >90% satisfactory recognition rate has been removed due to its source's unreliability.
2. Software sensitivity and technical aspects discussed in detail (Lines 356-368).
3. Explanation provided for similar accuracy across software versions (Lines 334-344).
4. Significance of arch form studies and clinical implications expanded (Lines 308-333).
5. Overall improvement with the incorporation of suggested references (Lines 1-396).
Conclusions (Lines 380-396): Revised to offer a cautious interpretation of findings, acknowledging study limitations
For the specified lines, please consult the attached document.

Round 2
Reviewer 3 Report
Comments and Suggestions for Authors
Dear authors,
Thank you for revising the manuscript according to my comments.
Reviewer 4 Report
Comments and Suggestions for Authors
The paper was increased by the authors.